# Peer review of "Harnessing Nanotechnology: Emerging Strategies for Multiple Myeloma Therapy"

_biomolecules, 2024, doi:10.3390/biom14010083_

Round 1

Reviewer 1 Report

Comments and Suggestions for Authors

The manuscript and the topic is of interest but contains mistakes

Content regarding immunotherapy in myeloma should include bispecific antibodies

The extension is too long and should be reduced and should focus mainly in nanotherapy in myeloma

Page 1  Line 43.  Reference should be in parenthesis

Page 2  Line 77. Reference should be in parenthesis

Text in Page 2   Lines  75 to 79

Text Page 3 Lines 87 to 106

Both paragraph regarding myeloma therapy are repeated, one of them should be removed

Page 3 The text below, regarding side effect of myeloma therapy in too simple and cantains mistakes: targeted therapy cause other side effects...mainly CART and bispecific antibodies

Chemotherapy, for instance, is often associated with nausea, fatigue, and neutropenia [19], while targeted therapies can cause 110 peripheral neuropathy and thrombocytopenia [20]

References are not enough updated

For example they should include this one:

Nanoparticle-Based Approaches for Treatment of Hematological Malignancies: a Comprehensive Review.

Hani U, Gowda BHJ, Haider N, Ramesh K, Paul K, Ashique S, Ahmed MG, Narayana S, Mohanto S, Kesharwani P.

AAPS PharmSciTech. 2023 Nov 16;24(8):233. doi: 10.1208/s12249-023-02670-0.

Reviewer 2 Report

Comments and Suggestions for Authors

This review article by Yang et al explores the recent advances in using nanotechnology to diagnose and/or treat multiple myeloma. The authors do a good job introducing the pathophysiology of multiple myeloma cancer cells, although they might add some details to the characteristics of the cancer cells as in markers that can be used to target nanoparticles loaded with different drugs, as indicated below in descriptions of the different nanoparticle or nanocarrier designs. The review then covers the issues with current drugs and how nanoparticles can not only increase efficacy but also drug stabilities and specificity.  The authors then cover the different design of nano-particle-based drug delivery systems, which leads to specifics of nanoparticles that specifically target myeloma cells. The authors cover important areas of the delivery systems, but there are multiple redundancies that need to be addressed as detailed below. Case studies are then reviewed, which provide some specific details of patient responses, which is excellent. Future perspectives are covered, although this section requires a more concise review as some sections are a bit long-winded. The conclusion section could also be more concise. Overall, this is an interesting review and certainly informative, but the following issues must be addressed or at least considered. Most importantly are the paragraphs that are repeated or redundant, which must be removed and the sections reorganized. There are also some grammatical issues that should be tackled but will be easy to fix.

 Line 43, there is a typo with the reference insertion (I think), as there is the "3" after the phrase "lifetime risk of 0.7%". Agree that this needs a reference. These are throughout the document – are these footnotes?

Lines 51-53, it might be helpful to provide a bit more detail on the transformation of B-cell progenitors as they mature into plasma cells, and at what stage during this developmental process the cells become cancerous.

Lines 61-64, is there evidence of the cytokines released by myeloma cells that alter osteoclast production? Would be good to briefly describe, to be thorough.

Line 77, end of sentence, is the "4" referring to the reference #4 or a footnote?

Lines 87-92, this paragraph is a repeat of lines 65-70, so it needs to be removed.

Lines 93-99, this paragraph is a repeat of that above for lines 73-79, so this entire paragraph needs to be removed.

Lines 100-106, this paragraph is a repeat of lines 80-86. Given that the above paragraphs are also repeated from the previous ones, this entire section, lines 65-106, needs to be overhauled - while the information is important, this reviewer is now confused as to the best order of these concepts (and associated references).

Section 2, this set of paragraphs (some of which can be combined as mentioned above) could be divided into two subsections, one for the use of nanoparticles for drug delivery and a second focused on nanoparticles for cancer imaging and diagnosis, each with their own header.

Lines 225-228, are there specific receptors or cell surface proteins on Myeloma cells that can be targeted by "active targeting" nanoparticles, thereby targeted drug delivery? This should be mentioned, and in fact these characteristics of myeloma cells should be included in the introduction.

Lines 237-241, what genes associated with MM could be targeted with gene therapeutic particles? This should be carefully described.

Lines 251-254, some details on the improved half-life of Doxil compared to the non-encapsulated drug would be helpful.

Line 283, issue with number at the end of sentence.

Lines 308-310, some details on the improved efficacy of liposomal forms of bortezomib would be helpful - what specifically was improved? This will help the reader appreciate the improvement, while also understanding the problems associated with such drug formulas as detailed in the subsequent paragraphs.

Line 311, what about the heterogeneity of myeloma? This could use some details, as this is critical to address with targeted therapies.

Lines 322-324, as mentioned previously, these general statements of "improved efficacy" or "reduced toxicity" would be better appreciated with some details of such improvements. This is included in the subsequent section on clinical studies, but some specific details of what the studies indicated for improvements would add enthusiasm to this review of such studies.

Line 327, how did the polymeric nanoparticle versions of bortezomib perform? What were the delivery details, or their specificity for myeloma cells? Without any details, these are too broad of statements and therefore lose their impact on understanding the improvements provided by nanoparticles to treat myeloma.

Line 332, again details of loading capacities and delivery would be appreciated.

Lines 345-349, these sentences are simply restatements of previous sentences that already mentioned the efficacy of liposomal encapsulated bortezomib, or polymeric forms. How do they compare to directed delivery? How is this paragraph different from that stated above in lines 319-329? Overall, this is redundant.

Lines 365-366, this is a repeat of the statement made previously about low toxicity of liposomal formulas with doxorubicin, which is redundant - what did the studies show?

Lines 374-375, how much drug can be encapsulated, and how does this affect sustained release? Again, some actual details here would provide a better appreciation of how the encapsulation of different drugs can improve efficacy.

Lines 382-383, this is just a repeat of lines 360-361.

Lines 395-397, the authors need to further develop the concepts of "multifunctional nanoparticles" or "stimuli-responsive systems" - what makes these unique, how are they designed, and what have results shown as to their efficacy? Some details of these interesting designs are important to add this review.

Lines 421-427, what markers can be used to target the nanocarriers? Some details are needed here.

Lines 428-463, here the authors give specific results of studies, which is exactly the approach that should be taken with the previous sections.

Lines 470-472, the authors should at least mention some of these markers, to give the reader some precise information as to what can be built into the nanocarriers to specifically target myeloma cells.

Section 4 on case studies, this section is much better developed, and very useful in understanding the specific responses of patients to the nanoparticles, including specific side-effects and challenges.

Section 5, Future Perspectives and Challenges, while this is an important section, it reads more like a continuous train of thought, rather than a concise review of future goals, so suggest some tightening of the sentences and paragraphs to provide a concise review. In particular, sections on the "omics" of myeloid biology, and that covering scalable syntheses of nanocarriers, could be more concise. This can also be applied to the Conclusion section (6), much of which is simply restatements of those provided in Section 5.

Comments on the Quality of English Language

Throughout the manuscript, there are several very short paragraphs (some with only one sentence) that could be combined with other paragraphs, so a careful review is warranted. More importantly, there are multiple sentences or entire paragraphs that are repeated word-for-word, and therefore the manuscript requires careful editing to not only remove these paragraphs but also make sure the organization is logical.

Round 2

Reviewer 1 Report

Comments and Suggestions for Authors

There is a mistake regarding use of biespecific in Myeloma

Blinatumumumab is not indicated in Myeloma and should be removed in the paragraphah. Teclistamab and Talquetamab should be included

Author Response

Dear reviewer,

Thank you for your comments, we have revised the manuscript accordingly.

Reviewer 2 Report

Comments and Suggestions for Authors

Review of revised manuscript by Yang et al.

The authors do a great job summarizing the transformation of B-cell progenitors as they mature into plasma cells into cancerous cells (lines 49-102), which is greatly appreciated. The only minor comment would be to combine the sentence in lines 73-75 with the following paragraph starting on line 76 – the information provided in the stand-along sentence refers to the monoclonal antibody produced by the myeloma cells, which is also covered in the next paragraph so they can be easily combined.

The edits to a) cytokines released by MM, b) targets on the surface of myeloma cells appropriate for nanoparticles, and c) genes associated with MM are well-written and appreciated. The description on the improved stability of Doxil is also noted. 

Additional edits are also appreciated, including descriptions of myeloma heterogeneity, specific details of nanoparticles with improved efficacy and reduced toxicity, loading capacities, etc. This section is nicely detailed and thorough. This reviewer notes the in-depth information provided for diagnostic imaging of MM, along with improved chemotherapeutics and immunotherapeutics with nanotechnology.

Finally, the Future Perspectives and Potential Challenges section plus Conclusions are much improved, in particular the concluding remarks. Nicely done!

Other minor suggestion:

Lines 511-516, these should be combined into one paragraph, as the separate sentence on safety of nanotechnology-based treatments are covered in the following sentences.
